# Pressure-Based Posture Classification Methods and Algorithms: A Systematic Review

Luís Fonseca [1], Fernando Ribeiro [1,2,*] and José Metrôlho [1,2]

1   Polytechnic Institute of Castelo Branco, 6000-081 Castelo Branco, Portugal; f.luis@ipcbcampus.pt (L.F.); metrolho@ipcb.pt (J.M.)
2   DiSAC—Research Unit on Digital Services, Applications and Content, 6000-767 Castelo Branco, Portugal
*   Correspondence: fribeiro@ipcb.pt

**Abstract:** There are many uses for machine learning in everyday life and there is a steady increase in the field of medicine; the use of such technologies facilitates the tiresome work of health professionals by either automating repetitive tasks or making them simpler. Bed-related disorders are a great example where tedious tasks could be facilitated by machine learning algorithms, as suggested by many authors, by providing information on the posture of a particular bedded patient to health professionals. To assess the already existing studies in this field, this study provides a systematic review where the literature is analyzed to find correlations between the various factors involved in the making of such a system and how they perform. The overall findings suggest that there is only a significant relationship between the postures considered for classification and the resulting accuracy, despite some other factors such as the amount of data available providing some differences according to the type of algorithm used, with neural networks needing larger datasets. This study aims to increase awareness in this field and give future researchers information based on previous works' strengths and limitations while giving some suggestions based on the literature review.

**Keywords:** posture; classification; lying; bedded; pressure; algorithms

## 1. Introduction

The use of machine learning (ML) techniques in health research is becoming more common nowadays and it has the possibility of alleviating health professionals' busy schedules. A popular use of these techniques made popular recently is posture classification; using different ML techniques, researchers intend to accurately classify a certain patient's lying posture to aid in sleep or other in-bed related conditions. Although there are some studies that show satisfyingly accurate results of such applications, there are some data that seem to relate to the accuracy of these sort of applications. Examples of these altering features are, for example, the dataset used for the training of the ML model chosen by the researchers and the number of postures the algorithm is capable of classifying, among others.

With the purpose of making a more informed study and to better understand which factors might lead to a more accurate classification system, research on the technologies involved in the subject of posture classification was carried out. In this research, the topics found most often were all related to ML, the data used for the training of the ML algorithm, what steps were taken to process that data, and most importantly what method was used and its accuracy.

The data used for posture classification studies usually includes pressure data; these are gathered in a few different ways but mostly with the use of pressure (piezoelectric) sensors. In this case, a matrix of sensors of varied dimensions is placed under a bedded subject and each of the sensors will output the pressure value; these values result in a sort of pressure image which is tagged with the actual posture the subject is in and can then be used in training different ML algorithms.

Despite some studies using these pressure data directly for classification method training, there are some extra pre-processing techniques that can not only improve the accuracy but also the performance of the resulting system. These techniques include centering the pressure image, which increases the similarity between the multiple images available in a dataset; the images can also be further processed by a feature descriptor such as HOG (histogram of oriented gradients). This technique will focus on the shape of an object and yields better results when compared to the use of raw images; it also lowers the computation costs for training and classification.

Probably the most important factor in a classification system is the algorithm performing the actual classification, and these algorithms have evolved even in the subject of posture classification with studies using $k$-nearest neighbors ($k$NN) algorithms all the way to state-of-the-art neural network algorithms. The latter have many variations and show an increase in accuracy when compared to simpler approaches such as $k$NN or naïve Bayes.

This review aims to find any work that uses pressure data for posture classification, regardless of the type of data or the methods found in the reviewed articles. For organizing not only the information obtained from the studies included in this work, but also for better conclusions, the following set of research questions (RQs) was set:

RQ1: What is the number of samples in the dataset used?

RQ2: Is the dataset used available publicly?

RQ3: What data are gathered and used for posture classification?

RQ4: How many postures are gathered in the dataset and how many of them are later used for posture classification?

RQ5: What methods are used in posture classification?

RQ6: What is the accuracy of the proposed methods?

Answering these questions will result in a more informed discussion as their answers are used together to compare how the different approaches affect the outcome of the studies included in this review.

The review was organized according to the Preferred Reporting Items for Systematic Reviews and Meta-Analyses (PRISMA) statement [1]. This process included identifying the purpose and the intended goals of the review, a literature search, setting the inclusion and exclusion criteria, data extraction and analysis, a discussion and conclusions, and writing of the review.

The remainder of this article is organized as follows. Section 2 describes a search carried out to find related works and presents an analysis of some works that have some similarities with the work presented in this article. Section 3 contains the description and application of the methodology chosen to perform this review, and the search strategy, inclusion and exclusion criteria, and results are presented. Section 4 provides the data extraction and data analysis, and Section 5 presents the discussion. Finally, Section 6 presents the final remarks, the strengths and limitations of this work, and a discussion about challenges and opportunities.

## 2. Related Works

Research in posture classification, namely using pressure data obtained using pressure sensors under a lying person, has attracted considerable attention in recent years. Currently, numerous studies propose distinct algorithms to approach this problem. This growing interest can be seen in the results obtained in some databases of scientific articles. Despite this, there are no studies that have presented literature reviews on algorithms for lying people's posture classification based on pressure data. As of March 2023, there were 174 articles retrieved from Scopus when the query "(lying OR bed*) AND (posture OR position) AND classification AND pressure" was searched through the following fields: document title, abstract, and keywords. With the same query, 109 articles were retrieved from the Web of Science database. Of these, approximately 52% of them were published in the last 6 years. However, the same research did not result in any work that has carried out a review of the existing literature. In this sense, this work represents a step

forward concerning other related works, thus representing a significant contribution to this study area.

Despite not finding an actual literature review on this subject, the works analyzed as this study was conducted included some comparisons between their methods and the ones published before theirs. For example, the authors in [2,3] analyzed multiple studies in some detail to illustrate how the differences in their approach change the accuracy of their results for the better when compared to similar works. In [4–6], the authors found some approaches that resembled the one developed in their work and compared them, highlighting aspects such as a higher number of classes predicted or simply a better result in accuracy. However, these comparisons tend to be biased to the purpose of the specific study as they intend to better in an already existing area instead of making an overall review of the existing literature as this work aims to do.

## 3. Methodology

This section contains a systematic review of studies/papers that address the use of one or multiple methods for posture classification, namely using pressure data obtained mostly using piezoelectric sensors under a lying person. The review was conducted according to the Preferred Reporting Items for Systematic Reviews and Meta-Analyses (PRISMA) statement [1]. Thus, the steps implemented for this review, resulting in the indicated sections were as follows:

1.   Identifying intended goals for the review (Section 1)
2.   Describing how the search was conducted (Section 3.1)
3.   Screening for inclusion (Section 3.2)
4.   Screening for exclusion (Section 3.3)
5.   Analysis (Section 4)
6.   Discussion (Section 5)

### 3.1. Search Strategy

Attempting to find the most results, for this review, there were three databases used, namely Scopus, Web of Science, and PubMed. The search terms were set considering the main goal of this review, and after analyzing the results using different terms, the ones that wielded the best results were chosen, focusing on posture classification carried out on any kind of pressure data. The resulting string for the search was then defined as follows:

(Lying OR bed*) AND (posture OR position) AND classification AND pressure.

Furthermore, the results were filtered to include all work published after 2013 up to 2023.

The search was conducted in March 2023, through the document title and keywords fields, and resulted in a total of 257 studies: 104 from Scopus, 80 from Web of Science, and 73 from PubMed (187 after removing duplicates).

### 3.2. Screening for Inclusion

The step following the result of the initial search was examining the studies by their title and abstract to filter which of them were to be analyzed further with the purposes of this review in mind or which ones were to be excluded. For this screening, the studies to include were the ones that met the following criteria: (1) studies that presented the use of one or multiple methods for posture classification, (2) studies that focused on bedded or lying subjects, (3) studies that detail the accuracy of the method used, (4) studies that were published in a scientific peer-reviewed publication, and (5) studies that were written in English. All studies that met these criteria were included for further analysis.

After this step, 147 studies were excluded (mostly for their titles containing pressure data that were not relevant to this study such as water pressure), leaving 40 studies.

### 3.3. Screening for Exclusion

The remaining studies were analyzed to assess whether they could be included in the in-depth analysis. For this step, each article was read to extract the classification methods used, what data they used, and posture-related information. Studies without enough information about the methods or algorithms used or the results of the application of the classification system were excluded. In this step, 18 studies were excluded. Four studies limited their classification to sitting or lying positions, another four used wearable sensors which did not result in pressure data, three of them used blood or lung pressure data, two were repeats of another study already present (the most relevant was kept), two did not contain information regarding the performance of the method applied, one focused on three-dimensional joint estimations, another used recorded images, and the last had no full text available. The resulting list consisted of twenty-two studies.

### 3.4. Results

The steps taken for the review methodology are represented in Figure 1, which indicates that after the literature search on multiple databases, 187 studies were obtained (after the removal of 70 duplicates), referred to as the 'identification' stage in the diagram; after application of the inclusion criteria identified in Section 3.2. 'Screening for Inclusion' and in the 'screening' section of the diagram, 147 studies were excluded, resulting in 40 studies. A full-text evaluation of the remaining 40 studies was performed, excluding 18 studies that did not meet the required criteria or did not match the focus of this paper; this stage is represented in the figure as 'eligibility'. The remaining 22 studies were the 'included' studies in the flow diagram.

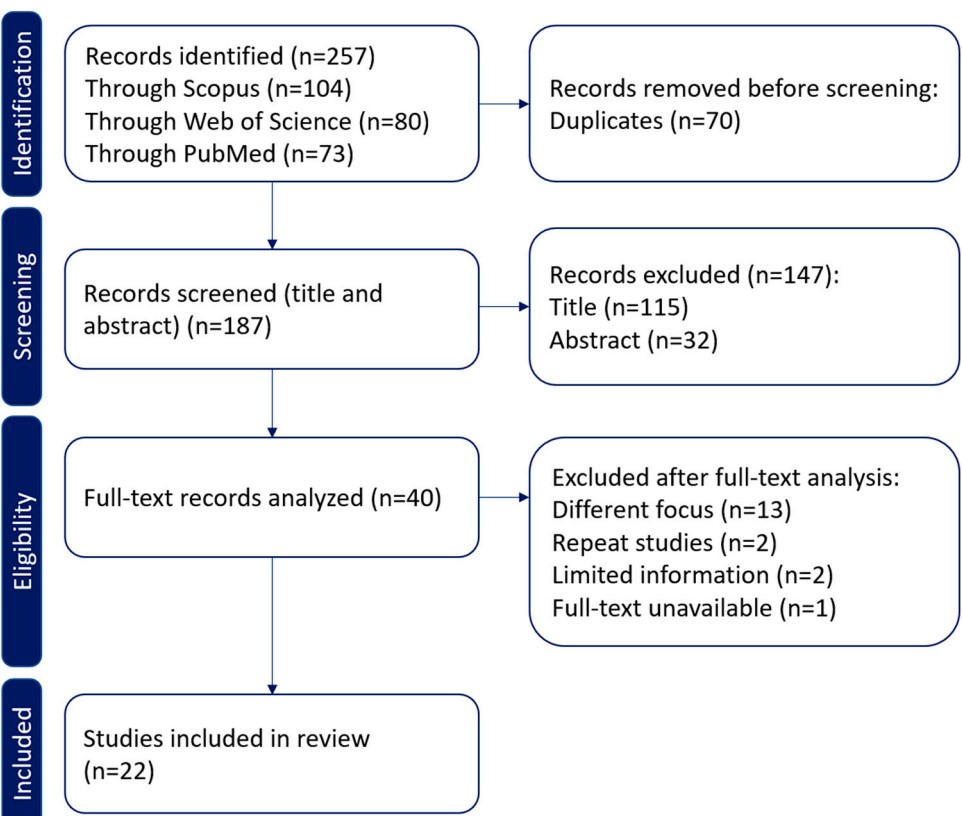

**Figure 1.** Diagram of the process that determined which papers were selected for this study (adapted from the PRISMA 2020 diagram [1]).

Most of the works selected for inclusion in this review use a dataset that includes a matrix of pressure values usually obtained by piezoelectric sensors; however, two studies were included with different approaches because the resulting classification method was similar to the ones that use pressure data. Although most of the studies examined throughout the screening steps used in-bed pressure data, a few works have high accuracy of different posture classifications with only a section of body pressure data and were also included.

*3.5. Characteristics of the Included Studies*

The studies chosen to be included in this review were published between 2013 and 2023. Although this was something that could be known from the initial query which limits the search to this range, it is still surprising that the studies that were included are distributed among the 10-year range.

The distribution portrayed in Figure 2 does, however, display an increase in the later years, with 50% of the studies having been published between 2021 and 2023.

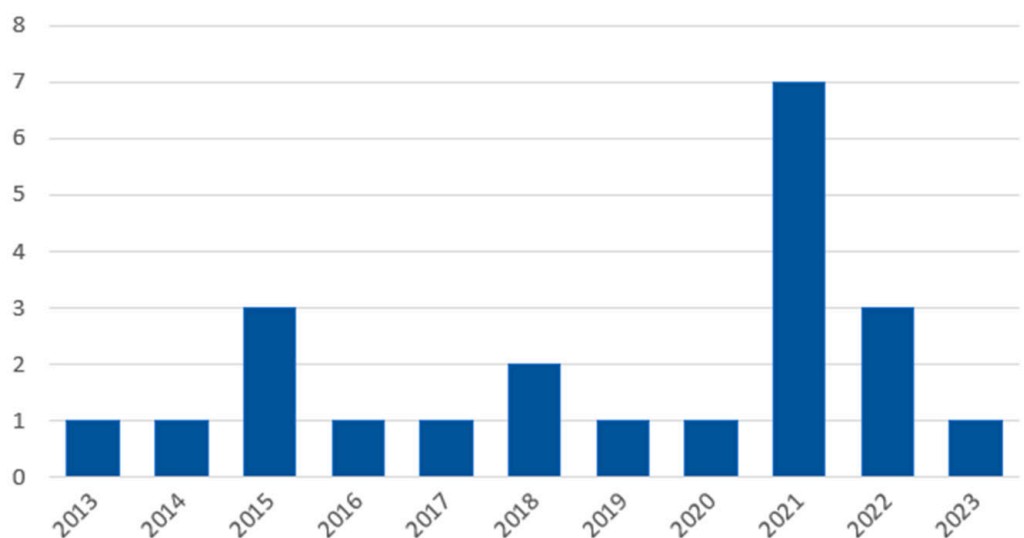

**Figure 2.** Publication date distribution for the studies included.

## 4. Analysis of the Included Studies

For easier analysis of multiple different studies, the most frequently found topics were found and withdrawn from the studies along with the information originally required to answer the research questions. These were then organized into a table including all the works selected for analysis. Each row represents one of the studies. The size column refers to the number of samples available in the dataset the authors used for their work; if this dataset is a publicly available one, the name of the dataset will also be included in the table cell. The data collected column has which data are included in said datasets. The poses available/used column displays how many postures the dataset originally has available followed by how many the researchers decided to use. The remaining columns refer to methods used either for pre-processing the data or actual classification and finally the accuracy of the developed system. Table 1 summarizes the data extracted from the selected articles.

**Table 1.** Synthetic analysis of the selected studies.

| Ref. | Size | Data Collected | Poses Available/Used | Pre-Processing | Methods/Algorithms | Accuracy |
|---|---|---|---|---|---|---|
| [4] | 26,000 PmatData | 32 × 64 pressure matrix<br>Age<br>Weight<br>Height | 17/3 and 17 | Median Filter<br>Histogram Equalization | Spiking neural networks | 99.99% (3 postures)<br>92.4% (17 postures) |
| [7] | 1440 | 1600 pressure array | 6/6 | N/A | Decision tree<br>Naïve Bayes<br>Support vector machine (SVM)<br>*k*NN | 84.5% to 96.8% |
| [2] | 3005 | 32 × 32 pressure matrix<br>Age<br>Weight<br>Height | 5/5 | Histogram of oriented gradients (HOG) | SVM<br>Convolutional neural network (CNN) | 84.8% to 91.24% |
| [8] | 2520 | 8 × 8 pressure matrix<br>Sex<br>Weight | 3/3 | N/A | CNN | 95.2% to 99.56% |
| [9] | 270 | 34 × 52 pressure matrix<br>Age | 9/9 | Center alignment | SVM<br>Naïve Bayes<br>Neural network<br>Random forest | 77.1% (highest) |
| [10] | 189 | 50 × 80 pressure matrix<br>Age | 6/6 | Feature extraction (not described) | SVM<br>CNN | 80%<br>70% |
| [11] | 2004 | 16 × 14 pressure matrix | 4/4 | HOG | SVM | 99.01% |
| [3] | 1116 | 64 × 27 pressure matrix<br>Sex<br>Age<br>Weight<br>Height | 4/4 | HOG<br>Local binary patterns | Feed-forward artificial neural network (FFANN) | 87.9% |
| [12] | 26,000 PmatData | 32 × 64 pressure matrix<br>Age<br>Weight<br>Height | 17/9 | Principal component analysis | *k*NN<br>Naïve bayes<br>FFANN | 94.9%<br>98.5%<br>99.6% |
| [5] | N/A | 32 × 64 pressure matrix | 8/8 | Gaussian lowpass filter<br>Binary filter | *k*NN | 97.1% |
| [13] | 448 | 80 × 40 pressure matrix<br>Sex<br>Age<br>BMI | 3/3<br>4/4 | N/A | Deep neural network | 99.7%<br>97.1% |

**Table 1.** *Cont.*

| Ref. | Size | Data Collected | Poses Available/Used | Pre-Processing | Methods/Algorithms | Accuracy |
|------|------|----------------|---------------------|----------------|--------------------|----------|
| [14] | 1051 | 32 × 32 pressure matrix<br>Sex<br>Age<br>Weight<br>Height | 4/4 | N/A | Deep neural network (ResNet-18) | 95.08% |
| [15] | 2004 | 32 × 16 pressure matrix<br>Weight | 4/4 | Bilinear interpolation<br>HOG<br>Scale-Invariant Feature Transform | SVM | 99.7% |
| [16] | N/A | 64 × 27 pressure matrix<br>Sex<br>Age | 13/3 | Binary image extraction<br>Center alignment | *k*NN | 98.4% |
| [17] | 736 | 30 × 11 pressure matrix<br>Weight<br>Height | 4/4 | N/A | Sparse representation | 91.4% |
| [18] | N/A | 32 × 64 pressure matrix | 5/5 | Median filter | Deep neural network with autoencoders | 98.1% |
| [19] | 26,000 PmatData | 32 × 64 pressure matrix<br>Age<br>Weight<br>Height | 17/3 | Reconfiguration of pressure maps into video files | Deep neural network (ResNet-18) | 99.8% |
| [20] | 26,000 PmatData | 32 × 64 pressure matrix<br>Age<br>Weight<br>Height | 17/17 | Median filter | Spiking neural network | 90.56% to 99.9% |
| [6] | 480 | 8 × 18 RFID matrix<br>Sex<br>Weight<br>Height | 8/8 | HOG | Decision tree | 96.14% |
| [21] | 26,000 PmatData | 32 × 64 pressure matrix<br>Age<br>Weight<br>Height | 17/5 | No description | *k*NN | 98.7% |
| [22] | 26,000 PmatData | 32 × 64 pressure matrix<br>Age<br>Weight<br>Height | 17/4 | N/A | Quantized fully convoluted neural network | 96.77% |
| [23] | 2076 | 13 × 15 pressure matrix<br>Breathing data | 4/4 | No description | Artificial neural network | 89.9% |

## 5. Discussion

With the analysis of the included studies concluded, the discussion will aim to relate the information found in the studies and all the factors involved in the classification methods proposed with their results. As with the analysis, this discussion will be organized according to the initial set of research questions, with each question having the possibility of affecting the outcome of the approaches in the studies included.

Regarding the amount of data used in each study and if they were obtained from a publicly available dataset (RQ1 and RQ2), six (27.3%) of the studies included using a publicly available dataset, namely PmatData [24] which includes the most (26,000) samples out of all the analyzed studies; the rest have their own data gathered with varying sample sizes (189–3005). The first noticeable difference is in the number of samples included in the studies' data. For algorithm training purposes, smaller datasets might lead to undertrained algorithms which will eventually lead to worse classification accuracy; the overall analysis of the work included in this review shows that the studies with smaller datasets have lower classification accuracy. However, the included studies do not always seem to have their results influenced by this matter. For example, the studies in [14,19] are similar in their method despite differences namely in the preprocessing stage and regarding the number of postures considered for classification; both show high accuracy (95.08% and 99.8%, respectively), with the first having a significantly smaller dataset to work on. This shows that the quality of the data has more influence on the classifier than the actual size of the dataset, as both datasets in these studies have about the same approach to the data gathering procedure, with [19] using the PmatData dataset. The fact that smaller data do not always lead to worse results has been studied in detail in [25], which displays the usage of different classification methods in the field of medical research on different-sized datasets and concludes that the performance of classifiers depends on how well the data represent the distribution and not on the amount of data available.

The quality of the dataset is dependent on what data are chosen to be included in it (RQ3), and with the focus of this review being pressure-related posture classification, all of the works include some kind of pressure image, with most using a matrix of pressure values acquired from piezoelectric sensors. The factors that might change the quality of the data are the dimension of the pressure image and what other information is gathered for each sample. Fourteen of the twenty-two (63.6%) studies have either the weight, the weight and height, or the body mass index (BMI) of the individual included in their data samples, but the accuracy does not seem to be directly influenced by the presence of body measurements as the studies that do not include any do not display a worse performance. This could be further explained by the possibility of the classifiers being able to predict these measurements, as the authors in [26] show by predicting the weight of a person using pressure maps as the input. Other characteristics such as age and sex are also gathered in some of the studies, but they do not seem to affect the outcome, and there is no seeming relationship noted between the dimension of the pressure maps and the accuracy either. This most likely indicates that the most important factor in pressure-related classification methods is the quality of the pressure data since all the studies using the public dataset have good classifier performance, and the ones that do not, vary in their results, even with some having higher resolution pressure images. This might be caused by excessive noise or misaligned data.

The postures considered for classification (RQ4) seem to be the factor most affecting the results in the studies, with some studies going to the extent of creating two different classifiers, with one considering a smaller number of postures, as in [4] where the authors end up with a significantly lower accuracy of 92.4% when 17 postures are considered compared to 99.99% accuracy based on only 3 postures. This fact might be related to the postures, as the postures considered usually include the four main postures found in most of the literature, and any posture beyond those is a variation of these, which are very similar postures that might be misclassified because of their similarity. However, the most probable reason is the number of classes, as with most classifiers, the higher the number of classes

the lower the accuracy of the classifier. This seems to have an influence on the studies as some have a low number of postures considered for classification despite having more postures available in the dataset used. However, the better classification rate might not be worth the risk of misclassification of postures on clinical usage, especially for research carried out in the medical field.

As this study looks for classification methods, the method or algorithm used in the actual classification (RQ5) is one of the most important aspects of the research if not the most important, but there are also methods included in some of the studies that are used to preprocess the pressure images mostly for two different reasons: reducing the computational cost of the resulting classification algorithm or improving its accuracy. The usage of preprocessing techniques such as HOG was, according to the authors in [2,6,11], an important factor in improving the accuracy of their classifier and was proven to do so by comparing their approach to similar ones.

Regarding the actual classifier, researchers in the studies included point out that the usage of neural networks has increased over time while other algorithms such as *k*NN, despite still being used, are mostly included for comparison; this is because, in the field of classification methods, neural networks have shown better performance. The dates on the publications depict that 9 of the 11 (81.8%) studies published between 2021 and 2023 use neural networks for their classifier while only 5 out of 11 (45.5%) used neural networks from 2013 to 2021. The research for this study shows that, for the most part, only neural network implementations reach a high of 99% accuracy, with the only exception being the work in [15] which uses SVM. It is also interesting that the lowest accuracy found [10] used a neural network for their classifier; however, the cause of the low accuracy can be related to the small dataset, as neural networks tend to have better results with a higher number of training samples.

As every different factor was assessed as to its influence in the resulting accuracy of the classification methods (RQ6), the data to answer this research question were used to relate to the other questions and to understand how they might affect the outcome of the included studies. Furthermore, there could be an analysis of the accuracies for all of the studies, but as they range from 70% to 99.99%, although the majority tend to land in the 90–100% range, there is not a definitive conclusion regarding the accuracy without comparing it to the other factors involved.

## 6. Final Remarks

This systematic review aimed to find various methods for pressure-based posture classification, and by following the steps suggested by the PRISMA methodology, the studies found more appropriate for this purpose were selected and reviewed in full to gather information and compare how the various factors involved in the development of such methods influence their results.

By conducting the search in three different databases (Scopus, Web of Science, and PubMed), a total of 187 distinct studies were obtained, which resulted in 22 included studies after applying different inclusion and exclusion criteria and full-text analysis when necessary. Despite the conclusions drawn from this review, the limited number of studies in this field does not allow for more definitive conclusions, and with only a few examples of each method available in the literature, the discussion findings in this review should still help future research in the field.

After the paper analysis and cross-referencing of the various factors involved in the studies included in this review, a few observations were taken for discussion and future reference. Regarding the data, the relationship between dataset size and accuracy does not appear relevant for most applications, and the need for additional data other than pressure maps does not seem to affect the outcome of the classifiers by much either. Furthermore, the datasets used by the researchers are not always similar, a more standardized data-gathering method should be considered, for example, by including the four main postures

found in the medical literature and by treating the rest of the postures as variations of the four initial postures.

The usage of preprocessing techniques should be considered as it is suggested to improve the computation requirements of the resulting classifier and in some cases to improve its accuracy; this relates to the importance of the pressure data as the preprocessing techniques focus on improving the pressure maps.

The factor that seems to influence the classification methods' accuracies the most is the number of postures considered for classification, and it has been noted by several researchers that it can also be easily explained as a machine learning issue: the more classes, the harder the classification. The studies have satisfying results even with a very low dataset size due to the reduction in the number of postures, with some studies even considering two of the postures mostly found in the medical literature (supine and prone) as one.

As machine learning evolves towards the use of neural networks, with deep neural networks being preferred, the sample number will be more important as deep neural networks tend to need more data to reach satisfying accuracy. The need for larger datasets will, however, address some of the issues described in the discussion, such as the posture numbers being a relevant factor for lower accuracy, as more data will allow researchers to use deeper neural networks and reach satisfying accuracy with a higher number of postures. The existence of more data would also lead to better comparison in the field, as most studies seem to attempt their use own data-gathering methods. If this was not the case and every study used the same amount of data or even the same dataset, comparing the results of their work would be far easier and more assertive conclusions could be taken.

Furthermore, as the pressure data were observed to be a determining factor in the outcome of the studies but the fact that it is not clear how pressure map dimensions affect them, it would be interesting to assess how the dimension would affect the resulting accuracy of the implemented methods. This way, a minimum optimal resolution could be found to lower not only the costs of the hardware (fewer sensors) but also the computational requirements for the classifiers.

**Author Contributions:** Conceptualization, L.F., F.R. and J.M.; methodology, L.F., F.R. and J.M.; validation, L.F., F.R. and J.M.; investigation, L.F.; writing—original draft preparation, L.F., F.R. and J.M.; writing—review and editing, L.F., F.R. and J.M. All authors have read and agreed to the published version of the manuscript.

**Funding:** This research received no external funding.

**Institutional Review Board Statement:** Not applicable.

**Informed Consent Statement:** Not applicable.

**Data Availability Statement:** Not applicable.

**Conflicts of Interest:** The authors declare no conflict of interest.

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
