# Peer review of "Pressure-Based Posture Classification Methods and Algorithms: A Systematic Review"

_computers, doi:10.3390/computers12050104_

Round 1

Reviewer 1 Report

The manuscript presented hereby show a literature review in order to assess the already existing studies on Pressure-based Posture Classification Methods and Algorithms. In this systematic review the literature is analysed to find correlations between the various factors involved in the making of  such a system and how they perform. The most important factors considered by the authors are:

- Publicly available data;
- Dimension of the dataset;
- Number and type of classes;
- Methods;
- Accuracy of the methods.

Overall the manuscript is well written and organised.

In my opinion the query string used for screening the initial documents could be complemented with different keywords such as "algorithms" or "machine learning" or "data mining". And probably this would result in a response of even more articles in recent years, leading to more accurate overview of the state of the art.

Despite this factor the articles used and covered are valid and make a good point on showing the main characteristics of this kind of work, meaning the overall goal of this paper is achieved.

One of the things authors should have considered is how the classes are  balanced within each of the data sets used. It seems correct the conclusion that some algorithms under perform with more classes but some of the studies provide only 5 or 6 classes and also have low accuracy. I believe this could be an interesting thing to add up to the table and to the discussion.

Author Response

We would like to thank the reviewers for their thoughtful and valued comments. In the following text (attached) we list all the reviews the manuscript and try to answer the questions placed by the reviewer. Our answer, in red, is presented after each question, referencing, when appropriate, the manuscript section containing the alteration. All the alterations performed to the manuscript are also in red for easy identification by the reviewer.

Reviewer 2 Report

This work presents a systematic review and analysis of pressure-based posture classification methods. However, it has several serious shortcomings that need to be addressed. The authors should consider revising their paper by addressing the following problems:

  1. The limited number of studies included in the analysis restricts the ability to draw assertive conclusions about the factors influencing posture classification accuracy. The authors should expand their literature search and include more relevant studies to strengthen their analysis.

  2. The varying data gathering methods used in the studies make it difficult to compare their results accurately. The authors should discuss the implications of this inconsistency and suggest a standardized approach for future research.

  3. The authors should consider citing the following works to strengthen their analysis and discussion: doi.org/10.1007/978-3-030-52067-0_22 and doi.org/10.1186/s40537-022-00586-3
  4. The impact of pressure map dimensions on classification accuracy is not adequately addressed in the work. The authors should investigate this aspect further and include their findings in the revised manuscript.

  5. The relationship between dataset size and classification accuracy remains inconclusive. The authors should provide a more in-depth analysis and discuss the role of data quality and representation in determining classification accuracy.

  6. The work does not provide a clear understanding of whether including supplementary information could improve classification accuracy. The authors should explore this aspect in more detail and provide a clear conclusion on its impact on classification outcomes.

  7. The relationship between posture numbers and classification accuracy is not fully understood. 

There are a few instances where the language could be improved for clarity and conciseness. Here are a few examples of suggested changes:

  1. Original: "Despite the conclusions withdrawn from this review the number of studies in this field is quite finite and does not allow for more assertive conclusions." Suggested change: "Despite the conclusions drawn from this review, the limited number of studies in this field does not allow for more definitive conclusions."

  2. Original: "Regarding data, it seems the relation between dataset size and accuracy does not seem to be relevant for most applications." Suggested change: "Regarding data, the relationship between dataset size and accuracy appears not to be relevant for most applications."

  3. Original: "The need for larger datasets will, however, solve some of the issues described in the discussion such as the posture numbers being a relevant factor for lower accuracy." Suggested change: "The need for larger datasets will, however, address some of the issues described in the discussion, such as the relevance of the number of postures as a factor for lower accuracy."

Author Response

(The authors gave the same response as above.)

Reviewer 3 Report

This paper makes a systematic review to address various methods for pressure-based posture classification, and the studies found more appropriate for this purpose are selected and reviewed in full of gathering information and comparing how the various factors involved in the development of such methods influence their results. After careful review, I consider this paper has an interesting topic, and it can be published after some minor problems are addressed in the revised manuscript, as follows:

1.      In Line 56, please check the “classification with studies using (kNN) k-Nearest Neighbors algorithms all the way to …”, and it should be revised for “(kNN) k-Nearest Neighbors”.

2.      In Lines 65-71, is what abbreviation for “RQ”, it is necessary to show out the full words of this abbreviation when it is in the first time of presentation.

3.      In Lines 143-147, it is lack for item (5).

4.      In Figure 1, for the “Records excluded (n=147): Title (n=115) Abstract (n=22)”, please check it because of the “115+22=137” not 147.

The English language can be improved by further checking, and I suggest this manuscript is good for proofreading.

Author Response

(The authors gave the same response as above.)

Round 2

Reviewer 2 Report

The authors answered to all my questions.